# Comparative Analysis of Dietary Habits and Obesity Prediction: Body Mass Index versus Body Fat Percentage Classification Using Bioelectrical Impedance Analysis

**DOI:** 10.3390/nu16193291

**Published:** 2024-09-28

**Authors:** Denisa Pescari, Monica Simina Mihuta, Andreea Bena, Dana Stoian

**Affiliations:** 1Department of Doctoral Studies, “Victor Babeș” University of Medicine and Pharmacy, 300041 Timisoara, Romania; denisa.bostina@umft.ro; 2Center for Molecular Research in Nephrology and Vascular Disease, “Victor Babeș” University of Medicine and Pharmacy, 300041 Timisoara, Romania; borlea.andreea@umft.ro (A.B.); stoian.dana@umft.ro (D.S.); 3Discipline of Endocrinology, Second Department of Internal Medicine, “Victor Babeș” University of Medicine and Pharmacy, 300041 Timisoara, Romania

**Keywords:** obesity, overweight, bioimpedance, adipose tissue, dietary habits, body mass index

## Abstract

**Background**: Obesity remains a widely debated issue, often criticized for the limitations in its identification and classification. This study aims to compare two distinct systems for classifying obesity: body mass index (BMI) and body fat percentage (BFP) as assessed by bioelectrical impedance analysis (BIA). By examining these measures, the study seeks to clarify how different metrics of body composition influence the identification of obesity-related risk factors. **Methods**: The study enrolled 1255 adults, comprising 471 males and 784 females, with a mean age of 36 ± 12 years. Participants exhibited varying degrees of weight status, including optimal weight, overweight, and obesity. Body composition analysis was conducted using the TANITA Body Composition Analyzer BC-418 MA III device (T5896, Tokyo, Japan), evaluating the following parameters: current weight, basal metabolic rate (BMR), adipose tissue (%), muscle mass (%), and hydration status (%). **Results**: Age and psychological factors like cravings, fatigue, stress, and compulsive eating were significant predictors of obesity in the BMI model but not in the BFP model. Additionally, having a family history of diabetes was protective in the BMI model (OR: 0.33, 0.11–0.87) but increased risk in the BFP model (OR: 1.66, 1.01–2.76). The BMI model demonstrates exceptional predictive ability (AUC = 0.998). In contrast, the BFP model, while still performing well, exhibits a lower AUC (0.975), indicating slightly reduced discriminative power compared to the BMI model. **Conclusions**: BMI classification demonstrates superior predictive accuracy, specificity, and sensitivity. This suggests that BMI remains a more reliable measure for identifying obesity-related risk factors compared to the BFP model.

## 1. Introduction

Currently, the increasing prevalence of chronic metabolic disorders, with overweight, obesity, and diabetes as primary factors, has become a significant public health concern in both developed and developing countries [1,2,3]. Obesity, recognized for its complexity and myriad associated complications, is regarded as a major public health issue. Various factors contribute to the early onset of these pathologies, including genetic predispositions, environmental variables such as pollution and sedentary lifestyles [4]. However, the most prevalent cause is the imbalance between excessive caloric intake and insufficient physical activity [5,6]. According to The Lancet, more than 1 billion people globally are expected to face obesity by the year 2022 [7]. Additionally, there has been a doubling in the number of overweight cases worldwide since 1990 and a fourfold increase in the number of children aged 5 to 19 affected by this condition [7]. In 2019, European countries reported that between 40–65% of their populations were diagnosed with overweight [8].

In addition to the global statistics, it has been observed that ethnic and racial minority groups present an additional risk of overweight and type 2 diabetes, thereby contributing to an increased risk of cardiovascular diseases [9,10,11]. The diagnosis of obesity is more frequent among women, with the highest increases reported among black populations [12]. Although all populations show an increase in the prevalence of obesity, the development of type 2 diabetes as a complication of obesity at a lower BMI (body mass index) value has been observed in Non-Hispanic Black (NHB) individuals compared to Non-Hispanic White (NHW) individuals [11,12,13,14]. This disparity is mainly attributed to significant differences in diet quality and eating habits [15,16,17].

Lately, there have been significant advances in understanding the pathophysiological mechanisms underlying the development of excess weight. Despite these substantial research efforts, these findings have yet to be widely applied in clinical practice, where obesity continues to be diagnosed and treated in a general manner [18]. The factors leading to obesity are multiple and interconnected, encompassing both genetic and environmental influences, with many being modifiable: food consumption, physical activity, and social and individual psychology [19,20,21]. Moreover, it has been observed that urbanization, wide access to high-calorie and refined foods, and a sedentary lifestyle are the main contributors to excess weight [1,21]. From a nutritional standpoint, excess weight has been correlated with various eating patterns, such as meal frequency, meal volume or quantity, types of snacks, the removal of breakfast as the main meal, and overall diet quality [22,23]. Among these factors, the most recognized and significant contributor to the current obesity epidemic is quantity of food intake, which disrupts the energy balance [24]. A diversity of foods and nutrients have been identified and associated with both the risk of obesity and the management of this pathology [25,26,27,28].

The daily and conscious eating habits of an individual, influenced by cultural and social factors, include the amount, type, and frequency of food consumed and are collectively defined as “eating patterns” [29,30,31,32]. Given that unhealthy food choices and, consequently, poor diet quality predominantly contribute to excess weight, identifying and analyzing food patterns is currently the most effective method for evaluating the influence of general eating routines among people with obesity [21]. Additionally, this approach allows for the association of dietary patterns with nutritional status variables, such as anthropometric measurements and cardiometabolic markers [33,34]. Increased portion sizes and the frequency of hypercaloric foods are the primary precursors of energy imbalance, and thus, obesity [30]. Accurately quantifying food patterns can be challenging due to differences in perception, including gender differences [35]. Moreover, assessing a person’s dietary pattern provides greater insights compared to evaluating daily individual macronutrient intake [36]. Investigating eating habits has transformed nutrition research by clarifying the complex relationship between food, nutrient interactions, and body composition. This has underscored the necessity for more detailed dietary analyses, especially in light of the global rise in obesity and related health risks. When discussing diet in its entirety, the terms “dietary patterns” or “food patterns” refer to the quantity, quality, and variety of foods and beverages consumed, as well as the frequency with which they are typically consumed [29,30]. However, dietary choices are influenced by a complex interplay of genetic, psychological, and environmental factors [4]. Over the past decade, portion sizes have significantly increased, leading to excessive calorie intake due to larger portions and frequent consumption of high-calorie foods. Research indicates that portion size is a key factor contributing to energy imbalance, weight gain, and ultimately the development of obesity [30]. Additionally, an increased risk of obesity has been strongly associated with more frequent consumption of breakfast away from home [37]. Moreover, studies suggest that individuals who skip breakfast tend to consume more calories overall throughout the day [23]. In recent years, it has become increasingly clear that evaluating dietary patterns provides a more accurate representation of daily food consumption than measuring macronutrient intake in isolation [36].

Although the report justifying the value of the most widely used anthropometric indicator, BMI, was proposed approximately 200 years ago, it was only 50 years ago that a concrete definition was implemented for this concept [38,39,40]. This delay resulted from prolonged debates in the field of research to confirm that BMI is a useful and accurate tool for diagnosing overweight and obesity [38,41]. Therefore, to evaluate the nutritional status of an adult, BMI is the most commonly used parameter, as it defines the excess of adipose tissue with generalized distribution [42]. Despite its widespread use, uncertainties persist regarding the relationship between BMI and mortality risk [41]. Nonetheless, studies have identified associations between BMI and both general and cardiovascular mortality risks, with stronger correlations observed among young adults and more significant associations in men compared to women [43,44]. A significant limitation of BMI is its inability to provide detailed information about the distribution and percentage of adipose tissue in the body [44,45,46]. Moreover, this parameter can provide misleading information in various situations, such as during different stages of life like childhood and adolescence, for performance athletes, or during the dynamic monitoring of a weight loss process [47]. Additionally, the percentage of adipose tissue varies by sex and age, regardless of BMI [45]. However, a universal definition of adult obesity based strictly on the percentage of adipose tissue has not been established [48,49]. In addition to BMI, other anthropometric parameters can be used to assess fat mass distribution both at baseline and over time in adults with excess weight. Two of the most commonly recognized measures are waist circumference (WC) and waist-to-hip ratio (WHR) [50]. Waist circumference (WC) is a well-established indicator, closely linked to visceral or abdominal fat mass, and serves as an effective predictor of potential complications associated with obesity [51,52]. Likewise, waist-to-hip ratio (WHR) is a crucial anthropometric measure, especially valuable for evaluating increases in insulin resistance [53]. The fat mass index (FMI) is a more precise indicator for evaluating body fat, but its interpretation is also limited [48,49,54,55]. Specific thresholds for FMI values have not been identified, unlike the percentage of adipose tissue used for diagnosing obesity [56,57]. Moreover, there are additional indices for quantifying adipose tissue mass. One of these is the body adiposity index (BAI), a useful and non-invasive tool [58,59]. However, it has been observed that in women with severe obesity, this parameter has not proven to be an adequate substitute for BMI [60]. Compared to other anthropometric indices, BAI has limitations in clinical practice and should be determined separately based on gender [61]. Furthermore, it has been found that BAI is not a universally applicable index that could replace BMI [62]. Moreover, there exists a broad range of indices for determining nutritional status. Among these are: Rohrer’s index, relative body mass index (RBMI), or BMIFat. The RBMI is an adjusted form of BMI that accounts for age and sex-specific variations, providing a more accurate assessment of body composition across different populations. It has been used in studies to improve the precision of nutritional status evaluations, particularly in populations where traditional BMI may not adequately reflect health risks [63,64]. Rohrer’s index, also known as the ponderal index, is a body composition measure calculated by dividing body mass by height cubed (kg/m^3^). It is particularly useful in pediatric and adolescent populations as an alternative to BMI, providing potentially greater sensitivity in distinguishing between lean and adipose tissue mass. This index may offer clinical advantages in assessing body fat distribution, especially where BMI might not be as accurate [65,66,67].

Conversely, the percentage of adipose tissue determined by electrical bioimpedance techniques can fall within optimal limits based on age, sex, and ethnicity, as indicated by the device-specific reference charts for both adults and children [68,69,70]. On the other hand, it has been found that BMI compared to body fat percentage is an inadequate screening tool for assessing body fat, particularly in individuals with mild adiposity. Therefore, it has been suggested that body fat quantification should be considered in the clinical evaluation of patients with obesity [71]. Furthermore, significant differences between the two models were observed in predicting risk factors associated with metabolic syndrome in patients with diabetes mellitus; however, the conclusion was that both models should be taken into consideration [72].

Obesity is known to negatively impact quality of life and life expectancy. However, the risk of developing complications varies significantly among individuals, a variation that cannot be fully explained by BMI or the degree of adipose tissue alone [73]. Metabolically healthy obesity (MHO) is a well-recognized condition, with a prevalence between 10–30%, and is influenced by factors such as gender and age [73,74]. While there is no standard definition of MHO, it is characterized by a lower cardiometabolic risk compared to metabolically unhealthy obesity (MHU) [75,76,77]. Despite the significant role of adipose tissue distribution in defining obesity subtypes, MHO remains a transient phenotype. Therefore, nutritional intervention aimed at weight loss is crucial for MHO individuals, as the presence of excess adipose tissue is more critical than its distribution when compared to normal-weight individuals [73].

Adipose tissue, due to its complexity and various functions—such as mechanical protection, lipid storage, thermogenesis, and regulation of systemic energy and nutrient homeostasis—is considered a metabolically active endocrine tissue with a remarkable ability to change size in response to different factors [45,78,79,80,81]. This dynamic nature of adipose tissue underscores the importance of comprehensive methods for evaluating body composition and health risks beyond BMI alone. This component belongs to the endocrine system, being essential in regulating homeostasis [82,83]. Consequently, its distribution, especially at the abdominal level, is a critical factor for the development of metabolic syndrome, type 2 diabetes and, implicitly, for the increase in cardiovascular risk [84].

With the advent of advanced technology, it is now possible to accurately determine not only the total amount but also the distribution of adipose tissue, both in percentage and in kilograms. Various devices have been developed and validated to assess the percentage of adipose tissue, including plethysmography, dual-energy X-ray absorptiometry (DXA), computed tomography (CT), and magnetic resonance imaging (MRI) [85,86]. However, the most commonly used technique in recent years is BIA [86]. Compared to other methods of quantifying body composition, BIA offers several advantages: it is easy to use, relatively low-cost, non-invasive, and can be applied to most individuals regardless of age [87]. These advantages make BIA a practical and accessible tool for widespread use in clinical and research settings.

Therefore, the aim of this study is to identify and analyze the impact of a wide range of eating habits and various medical conditions on body weight, as well as the potential risk factors contributing to obesity. This will be achieved using two different classifications of nutritional status: the BMI value and the percentage of adipose tissue, assessed through electrical bioimpedance. These models were constructed to identify and quantify the impact of various predictors on the probability of obesity. By contrasting the outcomes derived from the BMI-based and BFP-based models, this study seeks to highlight the similarities and differences in obesity risk factors as determined by these two classification systems. This comparative analysis offers a more detailed understanding of how different measures of body composition may influence the identification of obesity-related risk factors.

## 2. Materials and Methods

The prospective observational study was conducted over approximately four years, from July 2020 to June 2024, in our endocrinology unit. The total cohort comprised 1255 adults, 471 males (34%) and 927 females (66%), with a mean age of 36 ± 12, who were willing to assess their eating habits and undergo a comprehensive evaluation of their nutritional status with the aim of lifestyle modification. All participants provided informed consent. The study adhered to the ethical standards of the Helsinki Declaration and was approved by the Scientific Research Ethics Committee (CECS) of the “Victor Babeș” University of Medicine and Pharmacy Timișoara (No. 69/03.10.2022). To allow a complex comparative analysis of the final set of evaluated data, the entire group was divided as follows:

Based on the nutritional status evaluation, the cohort was divided into three subgroups according to BMI values [88,89,90]:**The control group** (BMI = 18.5–24.9 kg/m^2^), which consisted of individuals with no family or personal medical history of metabolic and cardiovascular diseases;**The overweight group**: BMI 25–29.9 kg/m^2^;**The obese group**: BMI ≥ 30 kg/m^2^.

Based on age and sex, independent of BMI value, according to the severity of excess adipose tissue [68,69], as follows:**Normal fat mass group (control group):**▪Normal fat mass women with distribution of adipose tissue according to age:20–39 years (21–32.9% adipose tissue);40–59 years (23–33.9% adipose tissue);60–79 years (24–35.9% adipose tissue).▪Normal fat mass men with distribution of adipose tissue according to age:20–39 years (8–20% adipose tissue);40–59 years (11–22% adipose tissue);60–79 years (13–25% adipose tissue).**High fat mass group:**▪High fat mass women with distribution of adipose tissue according to age:20–39 years (33–38.9% adipose tissue);40–59 years (34–39.9% adipose tissue);60–79 years (36–42% adipose tissue.▪High fat mass men with distribution of adipose tissue according to age:20–39 years (adipose tissue);40–59 years (adipose tissue);60–79 years (adipose tissue).**Very high fat mass:**▪Women with excess distribution of adipose tissue according to age:20–39 years (adipose tissue);40–59 years (adipose tissue);60–79 years (adipose tissue).▪Men with excess distribution of adipose tissue according to age:20–39 years (adipose tissue);40–59 years (adipose tissue);60–79 years (adipose tissue).


### 2.1. Patient Inclusion and Exclusion Criteria

**Inclusion criteria**: Participants were required to be adults, both men and women, with overweight or obesity, who have maintained the same residence for at least five years. They must have accurately completed a comprehensive questionnaire regarding eating habits, lifestyle habits, and personal and family history of cardiometabolic diseases. Additionally, a control group of normal-weight individuals without personal or family medical history of metabolic or cardiovascular conditions was included. Only those who consented to a thorough anamnestic and clinical evaluation by signing the informed consent form were included in the final assessment.

**Exclusion criteria:** Patients with secondary obesity, regardless of its etiology (e.g., endocrinological conditions such as hypothyroidism or Cushing’s syndrome; genetic disorders like Prader–Willi syndrome; or iatrogenic causes such as glucocorticoid or insulin therapy within the past 12 weeks) were excluded from the study [90,91,92]. Additionally, subjects who had followed a hypocaloric diet or received anti-obesity treatments (e.g., liraglutide, semaglutide, Orlistat, Bupropion/Naltrexone) within the past 16 weeks were excluded [90]. Individuals with documented psychiatric conditions and children were also not included in the research. Additionally, to minimize the risk of potential false-negative results from BIA, we excluded patients with certain conditions that could interfere with the accuracy of the measurements. Specifically, individuals with a history of neoplasms, including all types of cancers regardless of their origin or current status, were excluded from the study. Furthermore, participants with kidney damage were also excluded. This included those diagnosed with chronic kidney disease (CKD) at stages 3 to 5. Additionally, individuals with acute kidney injury (AKI) or other significant renal impairments were not considered for inclusion. Similarly, patients with liver damage were excluded. These included individuals diagnosed with chronic liver diseases such as cirrhosis, hepatitis B or C, or non-alcoholic fatty liver disease (NAFLD). These exclusion criteria were critical to ensuring that the bioimpedance analysis produced reliable and valid results, free from the potential confounding effects of these serious medical conditions.

### 2.2. Complete Patient Evaluations

Each participant was thoroughly informed about the stages of the study, including anamnesis and clinical and paraclinical evaluation, and they were provided with informed consent documentation. A detailed anamnesis was conducted for each patient, which encompassed the food survey, personal medical history of documented cardiometabolic pathologies, and family history, with particular attention to the presence of these conditions among first-degree relatives. Body analysis by bioimpedance was employed as the primary non-invasive method to estimate segmental body composition. Consequently, the following parameters were included in the initial evaluation of the participants:

**Body weight measurement:** Body weight was assessed using a mechanical scale with metrological certification, capable of measuring up to 200 kg. Participants were instructed to stand in a vertical posture on the scale while wearing minimal clothing.

**Height Measurement:** Height was measured using a calibrated wall-mounted stadiometer. Participants were instructed to stand in a vertical posture on the platform without wearing shoes.

**Nutritional status**: The nutritional status of each participant in our study was assessed using BMI, a widely utilized, cost-effective parameter. The BMI was calculated using the formula: BMI = weight (in kg)/height^2^ (in m^2^) [88,89], as previously mentioned.

**Family medical history and conditions**: A detailed family medical history focusing on cardiometabolic pathologies was obtained from each participant. The pathologies of interest, particularly concerning first-degree relatives, included obesity, overweight, type 2 diabetes, stroke, essential hypertension, and acute myocardial infarction.

**Personal medical history and conditions:** Similar to those mentioned previously, a pre-announced evaluation through directed questioning of the personal medical history was conducted individually for each participant. Moreover, for the documentation of these pathologies, each participant provided medical records during the initial consultation to verify their personal medical history. This process allowed for the exclusion of specific pathologies from the study. The primary pathologies emphasized were within the cardiovascular and metabolic categories, including diagnosed essential hypertension or use of antihypertensive treatment, prediabetes, type 2 diabetes, metabolic syndrome, lipid profile alterations, and asymptomatic hyperuricemia. Additionally, for patients who were undergoing insulin treatment in the past (more than 6 months), potential weight gain following its initiation was assessed. Among women, other parameters of interest included in the anamnesis were weight gain postpartum, diagnosis of polycystic ovary syndrome (regardless of subtype), daily use of oral contraceptives, onset of menopause, preclimax, and increased appetite during the premenstrual period. Regardless of gender and medical history, other significant factors addressed were previous weight loss, food intolerances, and weight gain after smoking cessation.

**Demographic and lifestyle factors:** In this category, parameters associated with the daily routine were also included due to the complexity of the anamnesis:oCigarette smoking status: this was defined as smoking at least one cigarette every day for more than a year.oAlcohol consumption: To quantify alcohol consumption, participants reported the number of units of alcohol consumed (equivalent to 10 mL of pure ethanol) via self-reporting. The units were defined as follows: two units equated to a pint or can of beer, one unit to a 25 mL shot of hard liquor, and one unit to a standard 175 mL glass of white or red wine. Participants consuming more than two units of alcohol daily were categorized as “alcoholic”, while those who had never consumed alcohol were classified as “non-alcoholic” [93].oPhysical activity level: To be excluded from the sedentary category, it was necessary to confirm a sustained physical effort of at least 30 min per day or 150 min per week (activity level > active plus basal).oSleep schedule: The duration of sleep for each subject was assessed, with a nightly duration of less than 7 h being classified as sleep deprivation or an insufficient sleep schedule [94].

**Eating habits and preferences:** The dietary habits of participants were meticulously documented from multiple perspectives. Key criteria included: daily breakfast consumption, adherence to the three main meals of the day, and the inclusion of two main courses at lunch. Additionally, the focus was placed on portion sizes relative to individual energy requirements, snacking between meals, the need for additional servings due to reduced satiety, and daily intake of fruits and whole foods. Eating habits were further categorized by quality, such as the consumption of home-cooked meals, dining at restaurants, fast-food intake, and dessert consumption. The study also examined the quantity of non-caloric clear liquids consumed, specifically plain water, as well as coffee and dairy consumption.

**Psychological and emotional factors:** Finally, psychological and emotional factors were assessed through targeted questions during the anamnesis, similar to the previous sections. The focus was on the presence of overeating or excessive eating triggered by negative and positive emotions, including eating for pleasure, as a reward, and secondary to loneliness, psychological stress, fatigue, or boredom.

### 2.3. Body Bioimpedance Analysis Variables

All subjects included in the study underwent an initial examination of their nutritional status using body bioimpedance analysis with the Tanita Body Composition Analyzer BC-418 MA III device (T5896, Tokyo, Japan). This analysis focused particularly on the percentage and distribution of adipose tissue. This involved a detailed analysis of the complete body composition utilizing a constant high-frequency current source (50 kHz, 500 μA) and employing a tetra-polar eight-point tactile electrode system. Participants were instructed to maintain an upright posture and grasp the analyzer’s handles to ensure contact with a total of eight electrodes, two for each foot and hand [95]. During the bioelectrical impedance analysis, a low-level electrical current passed through each participant’s body, and impedance (resistance to the current flow) was measured [96]. The entire procedure lasted approximately three minutes, and the results were thoroughly explained and recorded for each patient. Some studies have confirmed that in clinical settings, the Tanita Body Fat Monitor is accurate to within ±5 percent of the institutional standard for body composition analysis, DXA [97,98,99]. Tanita considers its method to be the most convenient and accessible for accurately predicting body composition [100]. The Tanita Body Fat Monitor Series produces repeatable results with a variation of within ±1 percent when used under consistent conditions [100]. Based on the results, the analyzed parameters were categorized into:Current weight or weight at the time of examination (kg);Metabolic basal rate (BMR) (kcal);Percentage of adipose tissue (%);Percentage of lean mass or muscle tissue (%);Percentage of total body water or hydration status (%).

The following personal information was collected and entered into the operating system of the instrument model used: identification data, gender, birth date, and height (cm). Upon entering the personal data for each patient, the instrument’s operating system automatically generated the BMR values.

### 2.4. Statistical Analysis

Numerical variables, based on their distribution type, were presented as median and interquartile range, while categorical variables were presented as frequency and proportions. The normality of distributions was assessed using the Shapiro–Wilk test, with a *p*-value < 0.05 indicating a non-Gaussian distribution. The test indicated non-normality for all numerical variables; hence, non-parametric methods were employed. To investigate differences between numerical variables, the Mann–Whitney U-test was used. For exploring statistically significant differences between categorical variables, the Pearson chi-square test was employed.

To identify the risk factor for both, overweight and obesity, we employed multinomial logistic regression. To assess the performance of the model, we used the Nagelkerke R^2^, which measures the proportion of variance explained by the predictors. Additionally, the Akaike Information Criterion (AIC) was applied as a criterion for model selection, helping to identify the best-fitting model by balancing model complexity and goodness of fit. We also used a backward elimination method to select predictors, systematically removing non-significant variables to arrive at the most parsimonious and predictive model.

To identify risk factors for obesity, multivariate logistic regression was used. The Nagelkerke R^2^ was used to assess model quality. The ROC curve parameters, including specificity, sensitivity, accuracy, and AUC, were utilized to compare the two models. The results were presented in both tabular and graphical formats. The statistical analysis was performed using R version 4.3.0 (R Core Team, 2024), a language and environment for statistical computing provided by the R Foundation for Statistical Computing, Vienna, Austria. A *p*-value < 0.05 was considered statistically significant, with a 95% confidence interval.

## 3. Results

The data collected from these patients are comprehensive, encompassing bioimpedance measurements, demographic and lifestyle factors, family medical history, eating habits and preferences, and psychological and emotional factors, as well as specific conditions and health issues. The analysis began by classifying the patients into control, overweight, and obese categories using the body mass index system. Subsequently, the classification was repeated using the body fat percentage system. The results from these two classification methods were then compared to understand the differences and similarities in the determinants of weight status identified by each approach.

### 3.1. Numerical Variables

The analysis of the study population using both the BMI model and the adiposity-based model revealed significant differences in various health and lifestyle parameters between normal weight, overweight, and obese participants, and between normal, high, and very high FM, as detailed in Table 1 and Table 2. In the BMI model, the highest median age was observed among the obese group (38 years), while the overweight group had the lowest median age (31 years) (*p* < 0.001). Conversely, the adiposity model indicated a higher median age among both very high FM (37 years) and high FM (35 years) individuals compared to the control group (34 years) (*p* < 0.001). In both models, the sleep duration was significantly longer in the control group compared to the study group, with statistically significant differences (*p* < 0.001). Regarding water intake, the control group exhibited the highest daily consumption (2500 mL/day), whereas the overweight and obese groups had the lowest (1500 mL/day) (*p* < 0.001). Consequently, hydration status, as evaluated by electrical bioimpedance, demonstrated statistical differences among the three groups (*p* < 0.001) in both models. These findings suggest that, regardless of the classification method, individuals with optimal body weight demonstrated healthier body composition and more appropriate lifestyle habits, such as higher consumption of non-caloric clear liquids and longer sleep duration, compared to those who are overweight or obese.

### 3.2. Demographic and Lifestyle Factors

Using the BMI classification for nutritional status, the demographic and risk factor analysis reveal statistically significant differences between the evaluated groups. The control group, characterized by a normative BMI, presents an even gender distribution (50% male; 50% female). In contrast, the overweight and obese groups show a significant skew towards a higher proportion of females, with a *p*-value of <0.001. This gender distribution pattern is consistent when comparing it to the body fat percentage (BFP) classification, where the group with high fat mass (FM) also has a higher proportion of females (76%) compared to males (24%).

Meal supplementation due to insufficient satiety is another area where significant differences emerge. In the BMI-based analysis, there is no reported supplementation in the control group, whereas 21% of the overweight group and 18% of the obese group report supplementation (*p* < 0.001). Similarly, under the BFP classification, supplementation is low in the normal FM group (3%) but increases to 17% and 20% in the high and very high FM groups, respectively.

A slight but significant difference is observed in holiday eating habits. Both classification models show increased food consumption during holidays, with the highest percentage in the very high FM group (15%) compared to 13% in the obese group based on BMI, indicating that holiday eating habits may be more closely associated with body fat percentage than BMI alone.

Television and device usage during meals present a stark contrast across groups. Under the BMI model, the control group has minimal TV meal consumption (2%), compared to 5% in the overweight and a substantial 88% in the obese group (*p* < 0.001). The BFP model echoes these findings, with a sharp increase in TV meal consumption from 4% in the normal FM group to 79% in the very high FM group, suggesting a strong correlation between increased body fat and sedentary behaviors.

Physical activity levels also significantly differ across groups. In both classification models, the control or normal FM groups engage in 30 min of daily physical activity at a much higher rate (94% in the control group and 82% in the normal FM group) than their overweight, obese, and high FM counterparts, where activity levels drop markedly (*p* < 0.001).

Alcohol consumption shows a clear trend of increasing with higher body mass and fat percentage. In both the BMI and BFP models, alcohol consumption is minimal in the control and normal FM groups but rises significantly in the overweight, obese, high FM, and very high FM groups, with *p*-values < 0.001 across the board.

Finally, smoking habits demonstrate an inverse relationship with weight and fat percentage. Smoking is more prevalent in the control group (39%) and the normal FM group (27%), declining significantly in the overweight, obese, high FM, and very high FM groups. This trend supports the hypothesis that lower body weight and higher metabolic rate are associated with smoking status.

These comparative results between the two evaluated models are detailed in Table 3 and Table 4, highlighting how different methods of classification (BMI vs. BFP) reveal nuanced variations in demographic and lifestyle factors.

### 3.3. Family Medical History

Table 5 and Table 6 present a detailed comparison of the family medical history of cardiometabolic pathologies based on BMI classification and body fat percentage (BFP) classification. The analysis reveals notable differences in the predisposition to obesity, type 2 diabetes, and cardiovascular diseases among first-degree relatives of participants across different groups.

In the case of family medical history (FMH) of obesity/overweight, a greater predisposition was observed among the control group when classified by BFP, with 24% of participants reporting a family history of obesity compared to only 9% under the BMI classification. However, the highest prevalence of FMH of obesity/overweight was found in the overweight group, with 99% under the BMI classification and 81% under the BFP classification, indicating a strong familial link to obesity across both measurement models.

Regarding FMH of type 2 diabetes, significant differences were noted between the groups. In the BMI classification, 60% of the overweight group reported a family history of diabetes, while 37% of the obese group did so. The BFP classification revealed a similar trend, with the highest prevalence in the high FM group (43%) and slightly lower but still significant levels in the very high FM group (39%). These findings suggest that both BMI and body fat percentage are associated with a familial predisposition to type 2 diabetes, with the overweight and high FM groups showing the greatest risk.

The analysis of FMH of stroke showed no reported cases among the control group in the BMI classification, while the BFP classification recorded a small percentage (4%) in the normal FM group. Both classification models indicated an increase in the prevalence of stroke in first-degree relatives as the BMI and BFP increased, with the highest percentages observed in the obese and very high FM groups (23% in both classifications).

Essential arterial hypertension showed a direct correlation with both BMI and body fat percentage. The prevalence of FMH of hypertension increased from 4% in the control group (BMI classification) to 56% in the obese group, and from 10% in the normal FM group to 55% in the very high FM group (BFP classification). These differences were statistically significant (*p* < 0.001), underscoring the link between increased adiposity and a familial predisposition to hypertension.

Lastly, the FMH of acute myocardial infarction (MI) displayed similar trends across both classification methods. The prevalence of MI in first-degree relatives was consistent between the overweight and obese groups in the BMI classification (11%) and the high and very high FM groups in the BFP classification (12%). This consistency suggests that familial risk for MI is similarly captured by both BMI and body fat percentage metrics.

These findings highlight significant variations in the family medical history of cardiometabolic conditions when considering different measures of body composition. The BMI and BFP classifications reveal that higher body mass and fat percentage are associated with a greater familial predisposition to obesity, type 2 diabetes, hypertension, and other cardiovascular conditions. These results emphasize the importance of considering both BMI and body fat percentage in assessing genetic and familial risks in overweight and obese populations.

### 3.4. Eating Habits and Preferences

Table 7 and Table 8 provide a comparative analysis of food preferences and habits among control, overweight, and obese groups based on both BMI values and body fat percentage (BFP). The findings indicate significant differences in dietary behaviors across these groups, emphasizing the impact of both BMI and adiposity on eating patterns.

Craving eating behavior was observed to increase with higher BMI and BFP levels, with only 1% of normal-weight individuals (BMI classification) and 7% (BFP classification) reporting this behavior. The prevalence of craving increased dramatically among the overweight and obese groups in both classifications, with 91% of obese individuals (BMI classification) and 87% of those with very high FM (BFP classification) exhibiting this behavior, highlighting a strong correlation between increased adiposity and craving eating habits.

Fast eating and large meal portions were less common in the control group, with just 1% of individuals in the BMI classification reporting these habits, compared to 13% and 15%, respectively, in the BFP classification. The prevalence of these behaviors increased significantly among the overweight and obese groups, with up to 94% in both BMI and BFP classifications. These findings suggest that both BMI and body fat percentage are strongly associated with the tendency to eat quickly and consume larger portions.

Daily fast food consumption showed similar patterns across both classifications in the control group (4% in BMI vs. 6% in BFP). However, significant differences were observed in the overweight group, where 14% (BMI) versus 34% (BFP) reported consuming fast food daily. Among the obese group, fast food consumption was notably high, with 85% (BMI) and 79% (BFP) reporting this behavior, indicating a consistent link between higher body mass/adiposity and fast food consumption.

Eating away from the table was not reported by any normal-weight participants in the BMI classification, whereas 5% of the normal FM group in the BFP classification reported this habit. This behavior was significantly more common among overweight and obese groups in both classifications, with up to 83% of obese individuals (BMI) and 79% of those with very high FM (BFP) eating away from the table, suggesting a shift towards more informal eating practices as body weight increases.

Dessert consumption after main meals was significantly higher in the overweight and obese groups compared to the control group in both classifications. In the BMI classification, 29% of the control group reported consuming desserts, compared to 87% in the obese group. The BFP classification showed a similar trend, with 33% of the normal FM group and 82% of the very high FM group indulging in desserts, highlighting the influence of increased adiposity on post-meal dessert consumption.

Snacking behavior was not reported by normal-weight individuals in the BMI classification but was common among overweight participants (*p* < 0.001). In the BFP classification, snacking was reported by 14% of the normal FM group and increased to 94% in the very high FM group, further emphasizing the role of body fat percentage in influencing snacking habits.

The consumption of meals at restaurants and on weekends was significantly more frequent in the overweight and obese groups in both classifications. Normal-weight individuals reported higher adherence to home-cooked meals and regular three-meal-per-day patterns, with 87% in the BMI classification and 76% in the BFP classification adhering to this routine. Breakfast consumption was notably low among obese individuals, with only 3% (BMI classification) and 8% (BFP classification) reporting regular breakfast intake, indicating a strong association between higher BMI/BFP and skipping breakfast.

The consumption of fruits and whole foods also decreased with higher BMI and body fat percentage. In the control group, 66% (BMI classification) and 57% (BFP classification) reported daily fruit consumption, whereas only 7% of obese individuals and those with very high FM consumed fruits daily. Similarly, whole food consumption dropped significantly from 71% in the control group (BMI classification) to 9% in the obese group, and from 61% (BFP classification) to 11% in the very high FM group.

These findings underscore significant differences in dietary preferences and habits across BMI and BFP classifications. The data highlight the importance of considering both body mass and fat percentage when assessing dietary behaviors and their impact on weight gain and overall quality of life. The patterns observed suggest that unhealthy eating behaviors become more prevalent as BMI and body fat percentage increase, regardless of the classification method used.

### 3.5. Psychological and Emotional Factors

Table 9 and Table 10 present a comparative analysis of psychological and emotional factors influencing eating behaviors among individuals classified by BMI and body fat percentage (BFP). The findings reveal statistically significant differences between the control, overweight, and obese groups across both classification models, highlighting the complex relationship between emotional states and eating behaviors in relation to body weight and fat distribution.

Stress-related eating was notably absent in the control group under the BMI classification, with only 5% of individuals in the normal FM group (BFP classification) reporting this behavior. In stark contrast, stress was a significant stimulus for eating in the overweight and obese groups, with 94% of obese individuals in the BMI model and 86% of those with very high FM in the BFP model indicating stress-induced eating. This suggests that higher BMI and body fat percentage are strongly associated with stress as a trigger for overeating.

Similarly, fatigue-related eating behaviors were minimally present in the control group, with just 1% in the BMI classification and 4% in the BFP classification reporting this habit. Among overweight and obese groups, however, fatigue-related eating was significantly more common, with 64% of obese individuals in the BMI model and 56% of those with very high FM in the BFP model exhibiting this behavior. These findings point to a clear link between higher body mass, increased fat percentage, and the likelihood of eating in response to fatigue.

Nibbling between meals, another key emotional eating behavior, was reported by 1% of the control group in the BMI classification and 10% in the normal FM group under the BFP model. This behavior increased dramatically in the overweight and obese groups, with 91% of obese individuals (BMI classification) and 86% of those with very high FM (BFP classification) engaging in nibbling, indicating a strong correlation between increased adiposity and this form of emotional eating.

Boredom-induced eating also showed significant differences across groups, with just 1% of the control group (BMI classification) and 3% of the normal FM group (BFP classification) reporting this behavior. In contrast, 44% of obese individuals in both classification models cited boredom as a reason for eating, suggesting that boredom becomes a more prominent trigger for overeating as body weight and fat percentage increase.

Compulsive eating behaviors were absent in the control group (BMI model) and reported by only 6% of the normal FM group (BFP model). However, these behaviors were prevalent in the overweight and obese groups, with 92% of obese individuals (BMI model) and 86% of those with very high FM (BFP model) exhibiting compulsive eating, indicating a significant relationship between higher BMI/body fat and compulsive eating tendencies.

Reward-based eating was another behavior significantly more common in the overweight and obese groups, with 79% of obese individuals (BMI classification) and 76% of those with very high FM (BFP classification) eating as a form of reward, compared to 0% in the control group (BMI classification) and 7% in the normal FM group (BFP classification).

The analysis also reveals that eating for pleasure, loneliness, and anger or upset were all significantly more common among overweight and obese individuals in both classification models. For example, eating in response to anger or being upset was reported by 90% of obese individuals in the BMI classification and 89% of those with very high FM in the BFP classification, compared to just 14% of the normal FM group in the BFP classification.

The results from Table 9 and Table 10 underscore the profound impact of psychological and emotional factors on eating behaviors in individuals with higher BMI and body fat percentages. These findings suggest that as body mass and fat percentage increase, so does the likelihood of eating in response to various emotional stimuli. The significant differences observed across both BMI and BFP classifications highlight the importance of incorporating psychological counseling and interventions in the comprehensive treatment of obesity, addressing the emotional and psychological components that contribute to unhealthy eating behaviors.

### 3.6. Specific Conditions and Health Issues

Table 11 and Table 12 present a analysis of specific conditions and health issues based on BMI and body fat percentage (BFP) classifications reveals significant differences across various health parameters, emphasizing the risks associated with increased BMI and higher adiposity. Food intolerance was relatively uncommon across all groups, with a slightly higher prevalence in the control group (5%) under the BMI classification compared to the BFP classification, where differences were not statistically significant. This suggests that food intolerance is not strongly associated with BMI or adiposity levels.

Premenstrual syndrome (PMS) exhibited a clear pattern, with no cases reported in the control group under the BMI classification, while 14% of overweight and 11% of obese participants reported experiencing PMS. A similar trend was observed in the BFP classification, where PMS was more prevalent in the high FM (12%) and very high FM (11%) groups. These findings indicate a significant relationship between PMS and higher BMI or adiposity.

Weight gain after quitting smoking was reported at low levels across both classifications, with a consistent 3% of overweight and obese individuals in the BMI classification, and similar figures in the high and very high FM groups in the BFP classification. This suggests a modest association between weight gain after smoking cessation and increased adiposity or BMI.

Oral contraceptive use showed significant variation, with higher usage among overweight individuals (7%) compared to the control (2%) and obese (4%) groups under the BMI classification. The BFP classification reflected similar trends, highlighting a potential link between oral contraceptive use and increased adiposity or BMI.

Menopause was more commonly reported among obese participants (13%) in the BMI classification, and among those with very high FM (12%) in the BFP classification. However, while the differences were statistically significant in the BMI classification, they were not significant in the BFP classification, suggesting that menopause may be more closely related to BMI than adiposity.

Preclimax symptoms and polycystic ovarian syndrome (PCOS) were both more prevalent in individuals with higher BMI or adiposity. Preclimax symptoms were reported by 7% of obese individuals and by 6% of those in the very high FM group, with significant differences across both classifications. PCOS showed a similar pattern, with 3% of overweight and obese individuals and 3–4% of those in the high and very high FM groups reporting the condition.

Prediabetes and dyslipidemia were strongly associated with higher BMI and adiposity. Prediabetes was reported by 8% of obese individuals and a similar percentage in the very high FM group, with significant differences in both classifications. Dyslipidemia showed a stark increase in prevalence among obese participants (77%) and those in the very high FM group (67%), underscoring the link between elevated BMI, adiposity, and these metabolic conditions.

Arterial hypertension followed a similar trend, with a significantly higher prevalence among obese individuals (18%) and those with very high FM (18%), indicating a strong association between hypertension and increased BMI or adiposity.

History of weight loss was notably higher among overweight and obese individuals (36% and 40%, respectively) in the BMI classification, and similarly elevated in the high and very high FM groups in the BFP classification. This reflects the commonality of weight loss efforts among those with higher BMI or adiposity, despite the challenges of maintaining weight loss.

Gout was rare but more prevalent among obese individuals (4%) and those with very high FM (3%), with significant differences observed in both classifications. Metabolic syndrome was significantly associated with both obesity and higher adiposity, with 88% of obese individuals and 79% of those with very high FM reporting the condition.

In contrast, post-insulin therapy weight gain and weight regain after metabolic surgery were not significantly different across groups, indicating that these conditions were rare among the study participants. Postpartum weight retention was reported by a small percentage of overweight and obese individuals in the BMI classification and by those in the high and very high FM groups under the BFP classification, with significant differences observed.

The analysis highlights the significant health risks associated with higher BMI and increased adiposity, emphasizing the need for targeted interventions to manage these conditions and improve the quality of life for affected individuals.

### 3.7. Multinomial Logistic Regression Analysis of Risk and Protective Factors for Overweight and Obesity

Multinomial logistic regression was employed to examine the associations between various predictors and the likelihood of patients being classified as normal weight, overweight, or obese. A Nagelkerke R^2^ of 0.907 suggests that the model provides a strong fit to the data.

Several predictors emerged as significant risk factors for both overweight and obesity. Notably, increased fat tissue was positively associated with higher odds of being overweight (OR = 1.18, *p* = 0.018) and obese (OR = 1.24, *p* = 0.002). This suggests that individuals with higher fat tissue are more likely to belong to the overweight or obese categories, with the effect slightly stronger for obesity.

A family history of hypertension also markedly increased the odds of both overweight and obesity. Those with such a family history had an 11.7-time higher risk of being overweight (OR = 11.69, *p* = 0.008) and a 16-time higher risk of being obese (OR = 16.01, *p* = 0.003). This highlights a strong hereditary influence on weight status.

For obesity specifically, eating dessert was a significant risk factor, with individuals who consumed dessert being over 16 times more likely to be obese (OR = 16.42, *p* < 0.001). This relationship was not significant for overweight status (*p* = 0.632).

Similarly, alcohol consumption was associated with a significantly higher likelihood of being obese (OR = 7.56, *p* = 0.036), although its effect on overweight status was not statistically significant (*p* = 0.221).

Lastly, prediabetes strongly predicted the likelihood of obesity (OR = 23.30, *p* < 0.001), with prediabetic individuals being more than 23 times more likely to be classified as obese. However, prediabetes did not significantly affect the odds of being overweight (*p* = 0.311).

In contrast, several factors appeared to be protective, significantly lowering the odds of being overweight or obese. Hydration status was one such factor, with better hydration reducing the likelihood of being overweight by 24% (OR = 0.76, *p* < 0.001) and the odds of obesity by 32% (OR = 0.68, *p* < 0.001).

Consumption of whole foods also served as a strong protective factor. Those who consumed whole foods were 95% less likely to be overweight (OR = 0.05, *p* < 0.001) and 96% less likely to be obese (OR = 0.04, *p* < 0.001), underscoring the importance of whole-food diets in maintaining a healthy weight.

Cooked food consumption similarly showed a protective effect. Eating cooked food reduced the odds of being overweight by 93% (OR = 0.07, *p* < 0.001) and the odds of being obese by 95% (OR = 0.05, *p* < 0.001), further highlighting the potential benefits of such dietary choices.

Another notable protective factor was the sleep hours, was found to be a significant protective factor for both overweight and obesity. Individuals who reported longer sleep durations had significantly lower odds of being either overweight or obese compared to those with shorter sleep durations. Specifically, each additional hour of sleep reduced the likelihood of being overweight by 77% (OR = 0.23, 95% CI: 0.12–0.45, *p* < 0.001) and the odds of being obese by 77% (OR = 0.23, 95% CI: 0.11–0.45, *p* < 0.001).

Additionally, eating three meals per day was strongly protective against both overweight and obesity. Those who ate three meals daily were 87% less likely to be overweight (OR = 0.13, *p* = 0.004) and 99% less likely to be obese (OR = 0.01, *p* < 0.001). This suggests that a regular meal pattern may help maintain a healthy weight.

Finally, age was a significant protective factor for overweight status, with each additional year of age associated with an 8% reduction in the odds of being overweight (OR = 0.92, *p* = 0.006). However, age did not significantly influence the likelihood of being obese (*p* = 0.359). The results are presented in Table 13.

### 3.8. Risk Factors for Obesity

To assess the risk factors for obesity, logistic regression models were employed using both BMI and BFP as classification criteria, with the latter now appropriately categorized into normal fat mass, high fat mass, and very high fat mass (indicative of obesity). Patients with a BMI ≥ 30 kg/m^2^ and those meeting the BFP thresholds were classified into the obesity category. The logistic regression models were designed to identify risk factors specifically associated with obesity, with the non-obese group (comprising both normal weight and overweight individuals) serving as the reference category. This approach was chosen to directly address the factors contributing to the likelihood of being obese. These models were designed to identify and quantify the impact of various predictors on the likelihood of obesity, considering the specific characteristics of each classification system. By comparing the results from the BMI-based and the newly adjusted BFP-based models, this study aims to provide a clearer and more accurate understanding of the risk factors associated with obesity. This comparative analysis highlights the similarities and differences in the identification of obesity-related risk factors between these two measures of body composition, offering a more comprehensive perspective on how these factors might be influenced by different methods of assessing body fat.

The logistic regression model based on BMI highlighted several significant risk factors for obesity. Age, a non-modifiable risk factor, was positively associated with obesity, with each additional year increasing the odds by 8% (OR: 1.08, CI: 1.03–1.13, *p* = 0.004). Emotional and behavioral factors also correlated positively with obesity risk. Notable factors include eating with appetite (OR: 17.08; CI: 4.82–68.55; *p* < 0.001), fatigue (OR: 16.90; CI: 5.20–64.03; *p* < 0.001), and stress (OR: 42.07; CI: 12.70–172.12; *p* < 0.001). Additionally, snoring (OR: 4.37; CI: 1.19–17.04; *p* = 0.029), compulsive eating (OR: 32.93; CI: 10.16–128.04; *p* < 0.001), and fast-food consumption (OR: 10.20; CI: 3.54–31.95; *p* < 0.001) significantly increased obesity risk. Interestingly, some factors were associated with a lower risk of obesity, contrary to expectations. These include eating quickly or consuming large amounts of food (OR: 0.18; CI: 0.04–0.76; *p* = 0.025) and consuming large portions (OR: 0.06; CI: 0.01–0.40; *p* = 0.005), potentially indicating compensatory behaviors. Similarly, eating at restaurants (OR: 0.12; CI: 0.03–0.49; *p* = 0.003), feeling angry or upset (OR: 0.14; CI: 0.03–0.53; *p* = 0.006), experiencing premenstrual syndrome (PMS) (OR: 0.10; CI: 0.03–0.38; *p* = 0.001), and using other devices while eating (OR: 0.15; CI: 0.03–0.79; *p* = 0.024) were all linked to lower obesity risk. Conversely, the presence of snacks after meals and the addition of portions were significant risk factors for developing overweight conditions (OR: 26.98; CI: 8.29–106.49; *p* < 0.001). Table 14 and Figure 1 present all data derived from the logistic regression analysis based on the BMI model.

In the logistic regression model based on BFP, several significant risk factors for obesity were identified, as detailed in Table 15 and Figure 1. A family history of diabetes notably increases the likelihood of developing obesity (OR: 1.66; CI: 1.01–2.76; *p* = 0.046), compared to the previously model. Regarding eating habits, binge eating is a significant risk factor (OR: 1.78; CI: 1.06–3.03; *p* = 0.031). Interestingly, eating quickly or consuming large quantities appears to be protective (OR: 0.39; CI: 0.16–0.91; *p* = 0.032). Additionally, detrimental lifestyle habits, such as smoking (OR: 4.97; CI: 2.09–12.36; *p* < 0.001) and daily alcohol consumption (OR: 1.98; CI: 1.11–3.57; *p* = 0.022), are significant risk factors for excess adipose tissue. Moreover, eating while watching TV was identified as a strong risk factor (OR: 2.28; CI: 1.26–4.13; *p* = 0.006).

Several risk factors appear consistently in both the BMI and BFP models, underscoring their robust association with obesity. Hydration status is protective in both models (BMI—OR: 0.84; CI: 0.75–0.92; *p* < 0.001. BFP—OR: 0.86; CI: 0.77–0.95; *p* = 0.005). Family history of diabetes has contrasting effects, increasing the risk only in the BFP model. Rapid or large-quantity eating behaviors are protective in both models (BMI—OR: 0.18; CI: 0.04–0.76; *p* = 0.025. BFP—OR: 0.39; CI: 0.16–0.91; *p* = 0.032). Experiencing PMS is associated with a lower risk of obesity in both models (BMI—OR: 0.10; CI: 0.03–0.38; *p* = 0.001. BFP—OR: 0.38; CI: 0.17–0.83; *p* = 0.017). These common factors highlight the complex interplay between physiological, behavioral, and emotional determinants of obesity risk. Figure 1 illustrates the differences between the logistic regression models for the two evaluated classifications.

### 3.9. Comparison between BMI and BFP Models

The comparison of the BMI and BFP models reveals differences in their predictive performance for obesity. The BMI model demonstrates exceptional predictive ability with an AUC of 0.998 (95% CI: 0.996–0.999), indicating near-perfect discrimination between obese and non-obese individuals. It also shows high accuracy (0.983), specificity (0.985), and sensitivity (0.982), suggesting that it accurately identifies both true positives and true negatives. On the other hand, the BFP model, while still performing well, has a lower AUC of 0.975 (95% CI: 0.967–0.982), indicating slightly less discriminative power compared to the BMI model. The BFP model’s accuracy is 0.922, with a specificity of 0.937 and a sensitivity of 0.904. These metrics suggest that the BFP model, although effective, is less accurate in identifying obese individuals compared to the BMI model. Overall, while both models perform robustly, the BMI model shows superior performance in predicting obesity. The results are presented in Table 16 and Figure 2.

## 4. Discussion

The diversity of eating habits, lifestyle factors, psychological and emotional influences, and genetic components all contribute significantly to an individual’s nutritional status, impacting the distribution of adipose and lean tissue masses. Obesity remains a contentious topic, often criticized for the limitations in its identification and classification. The importance of understanding these factors is critical for developing effective obesity management strategies. Although BMI has long been used as a standard measure to classify individuals based on their weight relative to height, it fails to account for the distribution and composition of body fat, which are crucial for understanding the health risks associated with obesity [101]. Contrary to expectations, the present study underscored the importance of BMI determination in clinical practice among overweight and obese individuals. While recognizing BMI’s limitations, our findings highlight its utility in providing a quick, accessible, and general indication of obesity, which can be essential for initial screenings and epidemiological studies. The implications of these findings are particularly relevant for nutrition clinicians, offering valuable insights for enhancing nutritional counseling and long-term monitoring of the nutritional status of overweight individuals. This approach can aid in developing personalized intervention strategies that address both dietary and lifestyle factors contributing to obesity. Furthermore, the study emphasizes the necessity of a comprehensive dietary assessment to identify the underlying factors leading to excess weight. This assessment should be part of a multidisciplinary team approach, allowing for the treatment of potential causes of obesity from various perspectives. Additionally, our research highlights the impact of different obesity classification models on the nutritional status of adults, regardless of their body weight, demonstrating that both BMI and adipose tissue measurements should be considered for a more accurate evaluation and management of obesity.

Furthermore, alongside modifiable risk factors for excess weight, such as dietary habits, several other parameters contribute to weight gain in individuals. Age is a prominent, yet immutable risk factor for overweight, primarily due to the progressive decline in basal metabolic rate with advancing age and the concomitant reduction in physical activity levels [102,103]. Additionally, aging is associated with a redistribution of adipose tissue, favoring visceral fat accumulation over subcutaneous fat [104]. In our study, statistically significant differences were observed between the study and control groups in both the BMI-based and BFP-based models (*p* < 0.001). Notably, in the BFP-based model, an increase in age was directly proportional to the degree of adiposity, contrasting with the BMI-based model, where younger ages were predominantly in the overweight category. However, age emerged as a predictor for obesity onset exclusively in the BMI-based model.

Measurements were conducted under comparable levels of hydration to avoid misinterpretation, as BIA is particularly sensitive to total body water. This method was utilized to estimate total body water, fat mass percentage, muscle mass, and total body water [105,106]. In the context of its application in sports and medicine, the raw BIA variable of phase angle, representing the ratio of resistance to reactance, has gained prominence and is provided by certain BIA devices [106]. Numerous studies have demonstrated the reliability of both single-frequency and multi-frequency BIA instruments, concluding that BIA can serve as a substitute for DXA in the analysis of whole-body and segmental body composition in large populations [107,108]. Recently, the association BMR and muscle tissue has been confirmed, both identified through body composition analysis [3,109]. Sarcopenia, a decline in muscle mass, frequently coexists with excess weight, especially among the elderly, representing an age-related abnormality [3,109,110]. The reduction in BMR is linked to excess weight, while variations in hydration status or total body water have been observed in both children and individuals with obesity [111,112,113]. Additionally, eating habits have been shown to negatively impact the percentage of adipose tissue, BMR, and hydration status [114,115]. In the current study, significant differences were observed in the parameters evaluated by BIA among the normal weight, overweight, and obese groups in both the BMI-based and BFP-based models. The highest percentage of adipose tissue was found in the obesity group, while the highest percentage of muscle tissue was attributed to the control group. BMR was comparable between the obese and normal weight groups but significantly lower in the overweight group. However, predictors for the development of obesity were identified only in the BFP-model. Specifically, an increased percentage of muscle tissue was found to be protective, reducing the chances of obesity by 32%. Additionally, a higher percentage of total body water also proved to be protective. BMR showed a small but significant increase in the odds ratio per unit (OR: 1.01; CI: 1.00–1.01; *p* < 0.001).

Various factors, some more extensively researched than others, contribute to the development of obesity. Physical activity, defined as “any type of body movement performed by skeletal muscles that results in energy expenditure”, plays a critical role [116]. Insufficient physical activity is linked to the onset of obesity, reduced cardiovascular fitness from childhood, and the development of various chronic cardiometabolic conditions in adulthood [117,118]. Consequently, a sedentary lifestyle is recognized as a significant contributor to obesity [118,119]. Additionally, regular physical activity has been associated with improvements in body composition and reductions in insulin resistance among adults [120,121]. Similar findings were observed in our study. In both evaluated models, the recommended physical activity of at least 30 min per day decreased proportionally with increases in BMI and adipose tissue percentage, being lowest in the group with obesity. The differences between the groups were significant (*p* < 0.001). Furthermore, a decrease in physical activity duration to less than 30 min per day was identified as a predictive factor for obesity development only in the BFP-based model.

Satiety after a meal, as well as food choices throughout the day, can be influenced by the availability of snacks. Although the term “snack” lacks a consistent definition, some studies associate it with factors such as the type and quantity of food consumed, the location, and the time of consumption [122,123]. Generally, snacks are linked to the consumption of healthy foods when there is a genuine feeling of hunger. However, in the absence of true hunger, snacks are more often processed foods that are high in calories and rich in saturated fats [124]. Chapelot et al. suggested that junk snacks contribute to weight gain [125]. In a cross-sectional study involving 1787 Norwegian adults aged 18 and over, Myhre, J.B., et al. found that workplace snacks tend to have a favorable nutritional profile, being a source of protein with lower sugar content compared to snacks consumed during holidays, at restaurants, or at home [126]. Additionally, a cross-sectional study of 958 Irish adults highlighted the nutritional role of snacks depending on where they were consumed, noting that eating in locations such as restaurants rather than at home is associated with larger portions and higher fat and fiber content [127]. While there is substantial evidence linking prolonged television viewing and reduced physical activity with the development of obesity and increased long-term cardiometabolic risk, other devices have not been conclusively associated with these outcomes [95]. However, there are insufficient data in the literature to definitively confirm the frequent consumption of snacks during television watching or the use of other devices as a significant contributor to these risks [128,129]. Interestingly, according to Chapman, C.D., et al., some television programs may positively influence lifestyle, as they found an increase in fruit consumption among individuals who watched less engaging or boring shows [105].

There are discernible differences in dietary patterns and eating behaviors concerning the quality and quantity of food by gender [130]. Gender-specific differences are also evident in the complications associated with obesity. A higher prevalence of obesity is typically observed among women, whereas men are more likely to develop metabolic complications secondary to excess weight [131,132]. Our study aligns with previous observations indicating that the prevalence of obesity is higher among women compared to men, with these gender differences being evident in most regions worldwide [133,134]. The type of adipose tissue expansion also varies by gender [133]. Although the overall obesity rate is 10% higher in women than in men, women tend to have a higher percentage of visceral fat [135,136,137]. In the current study, a significantly higher percentage of women was observed in the overweight (79%) and obesity (68%) groups within the BMI-based model. Similarly, in the adiposity-based model, 76% of the overweight group and 65% of the obese group were women. These findings indicate a higher prevalence of excess weight among females, with statistically significant differences (*p* < 0.001). However, gender was not identified as a risk factor for the development of obesity in the logistic regression analysis.

An association has been observed between excess weight and functional as well as structural changes in the brain’s reward system [138,139]. The consumption of hypercaloric and palatable foods in excess can trigger reward phenomena [100]. Eating behaviors and preferences vary according to gender and age [133,140]. Cross-sectional studies have shown that women consume more fruits compared to men [140,141]. Additionally, women tend to prefer sweet and easily accessible snacks such as candies, whereas men are more likely to opt for fast food items like pizza [142]. It has also been observed that men tend to eat more quickly, while women may eat uncontrollably, even in the absence of hunger [143]. These differences highlight the need for gender-specific approaches in dietary interventions and obesity management. In the current study, various dietary behavioral factors were identified as predictors of weight gain and obesity, with both similarities and differences observed between obesity classification based on BMI and that based on the percentage of adipose tissue. This comparative approach provided a more comprehensive understanding of how different body composition measures can influence the assessment and identification of obesity risk factors. Specifically, daily consumption of fast food, snacking immediately after the main meal, nibbling, and experiencing cravings significantly increased the risk of obesity in the BMI-based model. Conversely, behaviors such as quickly eating, dining in social settings like restaurants, and consuming larger portions were identified as compensatory behaviors and were not associated with an increased risk of obesity in the same model. All these eating behaviors showed statistically significant differences between the evaluated groups (*p* < 0.001). In contrast, the adiposity-based model identified different predictive factors for excess of adipose tissue. For instance, watching TV during meals increased the risk of obesity, while daily fruit consumption was a protective factor. Additionally, eating quickly and on the go did not predict the development of obesity in either model.

Both diet-related and psycho-affective factors play crucial roles in maintaining quality of life and preventing chronic diseases. Emotional eating, a problematic eating pattern, often negatively impacts eating decisions and is associated with various degrees of obesity and the emotional factors underlying its etiology [144]. Adult women are most frequently affected by emotional eating [145,146], which is linked to both psychological state and nutritional status [147]. Emotional eating results from an accumulation of emotions and behaviors associated with eating, rather than being considered a separate eating disorder [147,148]. The amplification of negative emotions plays a significant role in the onset and progression of obesity, demonstrating a bidirectional correlation between these entities [149,150]. Through a detailed anamnesis, our study identified psycho-emotional predictive factors for the development of obesity. Significant differences were observed between the normal weight, overweight, and obesity groups in terms of stress, fatigue, compulsive eating, reward, and pleasure. Notably, most of these factors were risk factors for obesity only in the BMI-based model. Eating for pleasure was the only predictive factor within the adiposity-based model, while upset/anger eating was protective in the BMI model but a risk factor for obesity in the adiposity-based model. These findings highlight the complex interactions between different models of body composition and their influence on the risk of developing obesity from distinct perspectives.

The current research also focused on identifying and analyzing parameters specific to females. In the BMI-based model, significant differences were observed among the control, overweight, and obese groups concerning menopause (*p* < 0.001). Conversely, in the BFP-based model, these differences were not significant (*p* = 0.15). Thus, menopause was identified as a risk factor for increased BMI but not for excess adipose tissue. However, the literature indicates that menopause is associated with changes in body composition, particularly an increase in the percentage of visceral adipose tissue, which amplifies cardiometabolic risk and deteriorates the quality of life among women [151]. This is primarily due to the absence of estrogen hormones during menopause, which significantly contributes to weight gain [152]. The reduction in circulating estrogens is also associated with a redistribution of adipose tissue, decreasing subcutaneous fat and increasing abdominal fat [153]. Additionally, it has been observed that the rate of developing obesity during menopause is three times higher compared to the pre-menopausal phase [154]. Therefore, in the present study, statistical differences were reported between the evaluated groups concerning pre-menopause diagnosis. Thus, pre-menopause was not noted as a risk factor for increased adiposity or BMI. Further research is necessary to clarify how hormonal fluctuations during menopause influence fat distribution, metabolic rate, and overall weight gain. Additionally, the synergistic effects of aging and menopause on these processes warrant more extensive investigation to inform the development of targeted interventions aimed at mitigating the associated health risks. Excess weight and the use of COCs are independent cardiovascular risk factors. Their concurrent use among women significantly increases the risk of pulmonary thromboembolism, with risk estimates ranging from 12 to 24 times higher [155]. In the current study, the use of COCs was not identified as a risk factor for the development of obesity. However, differences were observed between the groups; the overweight group had the highest percentage of COC use (7%) in both evaluated models.

The duration of sleep plays a crucial role in predicting cardiometabolic risk. Adequate sleep is crucial for maintaining cardiovascular health and metabolic function, as insufficient sleep has been associated with an increased risk of hypertension, obesity, diabetes mellitus, and cardiovascular disease (CVD) [156]. Short sleep duration (less than 7 h per night) has been linked to an elevated risk of cardiometabolic diseases [157]. In particular, short sleep duration can lead to alterations in glucose metabolism, increased appetite, and reduced insulin sensitivity [158]. In our study, the normal weight group exhibited a significantly longer sleep duration compared to the overweight and obese groups in both evaluated models, with statistically significant differences (*p* < 0.001). However, a sleep duration of less than 7 h per night was not identified as a predictor for obesity in this analysis.

Both overweight and smoking status are significant factors for cardiovascular risk [159]. Quitting smoking reduces cardiovascular risk, but it is often accompanied by increased appetite and subsequent weight gain [160]. While weight gain is a common consequence of smoking cessation, the health benefits of quitting smoking far outweigh the risks associated with moderate weight gain. Nicotine, the primary addictive substance in cigarettes, increases metabolic rate. When smoking is discontinued, the metabolic rate decreases, leading to fewer calories being burned at rest [161]. Research indicates that quitting smoking results in an average weight gain of 4–5 kg within the first year, though this can vary widely among individuals [162]. Interestingly, despite the weight gain following smoking cessation, cardiovascular risk was still reduced compared to those whose weight remained constant [163]. In the present study, smoking was more prevalent among individuals with normal weight, confirming the relationship between lower weight, higher metabolic rate, and smoking status. Additionally, both the BMI-based and BFP-based models showed a significantly reduced percentage of weight gain after quitting smoking. However, smoking status, specifically smoking at least one cigarette daily for more than a year, was identified as a predictive factor for obesity when classified according to the percentage of adipose tissue.

Similar to smoking, alcohol consumption is another harmful factor within the category of lifestyle choices. The impact of alcohol consumption on nutritional status is influenced by several factors, leading to significant inter-individual variations [164]. Cross-sectional studies have found that alcohol consumption is not consistently associated with BMI, regardless of gender [165,166]. However, gender differences have been observed, with a stronger association between alcohol consumption and BMI among men. This disparity is primarily attributed to the quantity and type of alcohol consumed [166]. The current results identified significant differences in alcohol consumption between the control group and the overweight and obese groups (*p* < 0.001), with consumption being higher among obese individuals in both evaluated models. Furthermore, alcohol consumption was identified as a risk factor for excess adipose tissue in the BFP-model.

According to the results, several risk factors were identified in both models—namely, BMI value and percentage of adipose tissue. However, certain predictors were associated only within a single model. This indicates that both models complement each other in the comprehensive assessment of an adult’s nutritional status. Furthermore, the comparative analysis of these models highlighted important differences in their predictive performance regarding various behaviors and eating patterns. The BMI-based model demonstrated an almost perfect discrimination between individuals with and without obesity, with an accuracy of 0.983, a specificity of 0.985, and a sensitivity of 0.982. These findings suggest that while each model has its strengths, their combined use provides a more holistic understanding of obesity-related risk factors.

While previous research has frequently highlighted gender differences in body composition and obesity prevalence, our study did not find a significant association between gender and obesity. Several factors may contribute to this discrepancy. First, differences in study design, including sample size, population characteristics, and methods of assessing body composition and obesity, might account for the variations in findings. Second, cultural, socioeconomic, and lifestyle factors that vary across different populations can influence obesity differently in men and women, which may not be captured uniformly across studies. Additionally, the role of hormonal differences, which are often cited in gender-related obesity research, might be less pronounced in our study population or could interact differently with other variables considered in our analysis. Finally, it is possible that the specific context of our study, including geographic location and time frame, resulted in a unique interaction between gender and obesity risk factors, leading to the observed lack of significant association.

Despite the promising results, there are several limitations associated with this study. While no other study has approached this subject with such precision, these findings must be interpreted within the context of these limitations. This study does not account for ethnic, cultural, and regional differences in the evaluated parameters, which negatively impacts the generalizability of the findings to populations with different eating habits compared to the evaluated groups. Another limitation is the use of the bioimpedance body analyzer, a device that may not be accessible to all clinicians for evaluating nutritional status and body composition, unlike the more readily available measurement of BMI. Additionally, the study did not aim to correlate the scores obtained from validated food and eating behavior questionnaires but rather to identify possible predictive factors for obesity from the perspective of two different classifications in a significant group of subjects. Furthermore, eating habits and their impacts on nutritional status can change over time, an aspect that was not dynamically monitored in this study. Therefore, this research represents a preliminary step toward a more comprehensive approach in the field of nutrition and obesity research. While the study aimed to identify risk factors for obesity, it did not include a separate analysis for normal weight and overweight categories. The logistic regression model was structured to compare obese versus non-obese participants, reflecting the study’s focus on obesity risk factors specifically. Future research could explore stratified analyses to further delineate risk profiles across different weight categories.

## 5. Conclusions

Although BMI has limitations in identifying and quantifying adipose tissue in individuals with obesity, it remains a valuable tool for evaluating adult nutritional status. Our research highlights the need for a multifaceted approach to evaluating and managing overweight adults, emphasizing the importance of integrating various assessment strategies. While the adiposity-based model showed promising sensitivity and specificity, the BMI-based model outperformed it in predicting obesity and overweight. Future studies should refine these models and explore their applications across different populations to enhance obesity prevention and treatment. This study underscores the importance of combining multiple evaluation methods to improve intervention strategies and public health outcomes.

## Figures and Tables

**Figure 1 nutrients-16-03291-f001:**
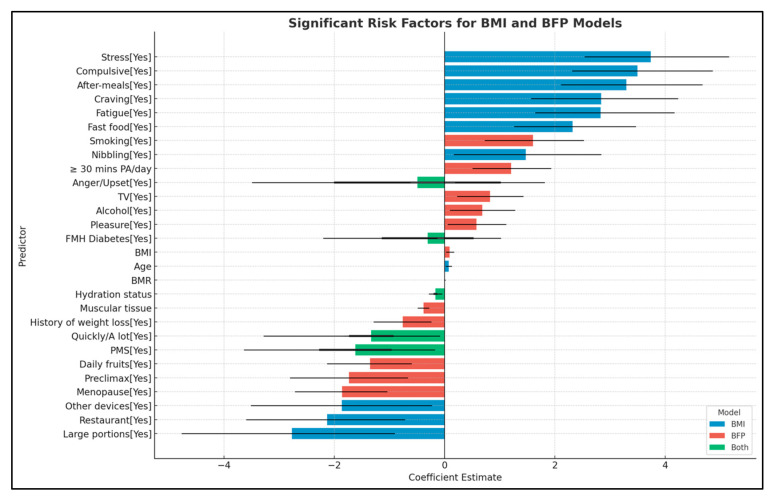
Significant risk factors for BMI and BFP models. Abbreviations: BMI—body mass index; BMR—basal metabolic rate; PMS—premenstrual syndrome.

**Figure 2 nutrients-16-03291-f002:**
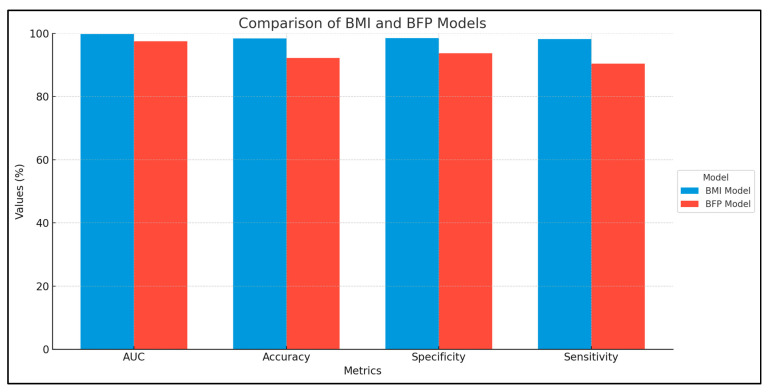
Comparison of BMI and BFP models.

**Table 1 nutrients-16-03291-t001:** Numerical variables based on BMI classification.

Variable	Control (*n* = 397)	Overweight (*n* = 261)	Obese (*n* = 597)	*p*-Value
Age (years)	33.00 (24.00–44.00)	31.00 (26.00–38.00)	38.00 (30.00–47.00)	<0.001
Water intake (mL)	2500.00 (2500.00–3000.00)	1500.00 (1000.00–2000.00)	1500.00 (1000.00–2000.00)	<0.001
Coffee (mL)	250.00 (170.00–300.00)	0.00 (0.00–200.00)	100.00 (0.00–200.00)	<0.001
Hours of sleep	9.00 (9.00–9.00)	7.00 (7.00–8.00)	7.00 (6.00–8.00)	<0.001
Adipose tissue (%)	21.00 (18.00–24.00)	35.00 (31.00–38.00)	42.00 (36.00–45.00)	<0.001
Muscular tissue (%)	76.00 (64.00–80.00)	61.00 (58.00–64.00)	54.00 (51.00–61.00)	<0.001
BMR (kcal/day)	1823.00 (1623.00–2139.00)	1498.00 (1404.00–1706.00)	1805.00 (1620.00–2199.00)	<0.001
Hydration status (%)	61.00 (57.00–63.00)	47.00 (44.00–50.00)	42.00 (39.00–46.00)	<0.001

Abbreviations: BMR—basal metabolic rate; *p*-value—Mann–Whitney U-test.

**Table 2 nutrients-16-03291-t002:** Numerical variables based on BFP classification.

Variable	Normal FM (*n* = 394)	High FM (*n* = 225)	Very High FM (*n* = 542)	*p*-Value
Age (years)	34.00 (27.00–44.00)	35.00 (29.00–43.00)	37.00 (30.00–46.00)	<0.001
Water intake (mL)	2500.00 (2500.00–3000.00)	1500.00 (1000.00–2000.00)	1500.00 (1000.00–2000.00)	<0.001
Coffee (mL)	250.00 (161.25–300.00)	0.00 (0.00–200.00)	100.00 (0.00–200.00)	<0.001
Hours of sleep	9.00 (9.00–9.00)	7.00 (7.00–8.00)	7.00 (6.00–8.00)	<0.001
BMI (kg/m^2^)	22.00 (20.00–23.00)	29.00 (27.00–31.00)	36.00 (32.00–40.00)	<0.001
Muscular tissue (%)	73.50 (64.00–80.00)	60.00 (58.00–62.00)	54.00 (51.00–60.00)	<0.001
BMR (kcal/day)	1795.00 (1558.00–2146.50)	1545.00 (1446.00–1788.00)	1801.00 (1598.00–2157.00)	<0.001
Hydration status (%)	60.00 (54.00–63.00)	46.00 (44.00–48.00)	41.00 (39.00–46.00)	<0.001

Abbreviations: BMR—basal metabolic rate; FM—fat mass; *p*-value—Mann–Whitney U-test.

**Table 3 nutrients-16-03291-t003:** Demographic and lifestyle factors based on BMI classification.

Variable	Class	Control (*n* = 397)	Overweight (*n* = 261)	Obese (*n* = 597)	*p*-Value
Sex	M	50%	21%	32%	<0.001
	F	50%	79%	68%	
Supplement	Yes	0%	21%	18%	<0.001
Holiday	Yes	8%	14%	13%	0.04
TV	Yes	2%	5%	88%	<0.001
Other devices	Yes	0%	7%	6%	<0.001
≥30 min PA/day	Yes	94%	16%	10%	<0.001
Alcohol	Yes	2%	26%	29%	<0.001
Smoking	Yes	39%	4%	7%	<0.001

Abbreviations: PA—physical activity; *p*-value—Pearson chi-square test.

**Table 4 nutrients-16-03291-t004:** Demographic and lifestyle factors based on BFP classification.

Variable	Class	Normal FM (*n* = 394)	High FM (*n* = 225)	Very High FM (*n* = 542)	*p*-Value
Sex	M	45%	24%	35%	<0.001
	F	55%	76%	65%	
Supplement	Yes	3%	17%	20%	<0.001
Holiday	Yes	7%	13%	15%	0.002
TV	Yes	4%	32%	79%	<0.001
Other devices	Yes	1%	3%	6%	<0.001
≥30 min PA/day	Yes	82%	16%	13%	<0.001
Alcohol	Yes	7%	20%	31%	<0.001
Smoking	Yes	27%	10%	9%	<0.001

Abbreviations: PA—physical activity; FM—fat mass; *p*-value—Pearson chi-square test.

**Table 5 nutrients-16-03291-t005:** Family medical history based on BMI classification.

Variable	Class	Control (*n* = 397)	Overweight (*n* = 261)	Obese (*n* = 597)	*p*-Value
FMH Obesity/Overweight	Yes	9%	99%	66%	<0.001
FMH Diabetes	Yes	4%	60%	37%	<0.001
FMH Stroke	Yes	0%	19%	23%	<0.001
FMH Hypertension	Yes	4%	49%	56%	<0.001
FMH MI	Yes	5%	11%	11%	<0.001

Abbreviations: FMH—family medical history; MI—myocardial infarction; *p*-value—Pearson chi-square test.

**Table 6 nutrients-16-03291-t006:** Family medical history based on BFP classification.

Variable	Class	Normal FM (*n* = 394)	High FM (*n* = 225)	Very High FM (*n* = 542)	*p*-Value
FMH Obesity/Overweight	Yes	24%	81%	69%	<0.001
FMH Diabetes	Yes	14%	43%	39%	<0.001
FMH Stroke	Yes	4%	16%	23%	<0.001
FMH Hypertension	Yes	10%	48%	55%	<0.001
FMH MI	Yes	4%	12%	12%	<0.001

Abbreviations: FMH—family medical history; FM—fat mass; MI—myocardial infarction; *p*-value—Pearson chi-square test.

**Table 7 nutrients-16-03291-t007:** Eating habits and preferences based on BMI classification.

Variable	Class	Control (*n* = 397)	Overweight (*n* = 261)	Obese (*n* = 597)	*p*-Value
Craving	Yes	1%	41%	91%	<0.001
Quickly/A lot	Yes	1%	82%	91%	<0.001
Large portions	Yes	1%	94%	94%	<0.001
Fast Food	Yes	4%	14%	85%	<0.001
Not at table	Yes	0%	20%	83%	<0.001
Dessert	Yes	29%	37%	87%	<0.001
After meals	Yes	1%	23%	95%	<0.001
Weekends	Yes	3%	18%	18%	<0.001
Restaurant	Yes	3%	8%	8%	0.003
Mindless	Yes	0%	83%	96%	<0.001
Childhood-like	Yes	0%	43%	92%	<0.001
Cooked food	Yes	74%	17%	16%	<0.001
Three meals/day	Yes	87%	43%	5%	<0.001
Breakfast	Yes	90%	44%	3%	<0.001
Two-course lunch	Yes	1%	57%	63%	<0.001
Daily fruits	Yes	66%	11%	7%	<0.001
Whole foods	Yes	71%	12%	9%	<0.001

Abbreviations: *p*-value—Pearson chi-square.

**Table 8 nutrients-16-03291-t008:** Eating habits and preferences based on BFP classification.

Variable	Class	Normal FM (*n* = 394)	High FM (*n* = 225)	Very High FM (*n* = 542)	*p*-Value
Craving	Yes	7%	52%	87%	<0.001
Quickly/A lot	Yes	13%	80%	90%	<0.001
Large portions	Yes	15%	87%	93%	<0.001
Fast Food	Yes	6%	34%	79%	<0.001
Not at table	Yes	5%	34%	79%	<0.001
Dessert	Yes	33%	47%	82%	<0.001
After meals	Yes	6%	46%	87%	<0.001
Weekends	Yes	5%	16%	18%	<0.001
Restaurant	Yes	4%	7%	8%	0.02
Mindless	Yes	14%	81%	94%	<0.001
Childhood-like	Yes	8%	56%	85%	<0.001
Cooked food	Yes	64%	22%	19%	<0.001
Three meals/day	Yes	76%	34%	11%	<0.001
Breakfast	Yes	80%	34%	8%	<0.001
Two-course lunch	Yes	11%	55%	61%	<0.001
Daily fruits	Yes	57%	13%	7%	<0.001
Whole foods	Yes	61%	14%	11%	<0.001

Abbreviations: FM—fat mass; *p*-value—Pearson chi-square.

**Table 9 nutrients-16-03291-t009:** Psychological and emotional factors based on BMI classification.

Variable	Class	Control (*n* = 397)	Overweight (*n* = 261)	Obese (*n* = 597)	*p*-Value
Stress	Yes	0%	20%	94%	<0.001
Fatigue	Yes	1%	8%	64%	<0.001
Nibbling	Yes	1%	54%	91%	<0.001
Boredom	Yes	1%	17%	44%	<0.001
Compulsive	Yes	0%	30%	92%	<0.001
Reward	Yes	0%	36%	79%	<0.001
Pleasure	Yes	0%	58%	44%	<0.001
Loneliness	Yes	0%	3%	58%	<0.001
Socially	Yes	1%	13%	12%	<0.001
Anger/upset	Yes	0%	85%	90%	<0.001
Major issues	Yes	0%	4%	87%	<0.001

Abbreviations: *p*-value—Pearson chi-square.

**Table 10 nutrients-16-03291-t010:** Psychological and emotional factors based on BFP classification.

Variable	Class	Normal FM (*n* = 394)	High FM (*n* = 225)	Very High FM (*n* = 542)	*p*-Value
Stress	Yes	5%	44%	86%	<0.001
Fatigue	Yes	4%	31%	56%	<0.001
Nibbling	Yes	10%	62%	86%	<0.001
Boredom	Yes	3%	16%	44%	<0.001
Compulsive	Yes	6%	47%	86%	<0.001
Reward	Yes	7%	42%	76%	<0.001
Pleasure	Yes	10%	41%	47%	<0.001
Loneliness	Yes	2%	24%	50%	<0.001
Socially	Yes	2%	12%	12%	<0.001
Anger/upset	Yes	14%	79%	89%	<0.001
Major issues	Yes	3%	29%	78%	<0.001

Abbreviations: FM—fat mass; *p*-value—Pearson chi-square.

**Table 11 nutrients-16-03291-t011:** Specific conditions and health issues based on BMI classification.

Variable	Class	Control (*n* = 397)	Overweight (*n* = 261)	Obese (*n* = 597)	*p*-Value
Food intolerance	Yes	5%	2%	3%	0.02
PMS	Yes	0%	14%	11%	<0.001
Weight gain after quit smoking	Yes	0%	3%	3%	0.005
Oral contraceptives	Yes	2%	7%	4%	0.004
Menopause	Yes	8%	3%	13%	<0.001
Preclimax	Yes	1%	5%	7%	<0.001
PCOS	Yes	0%	3%	3%	0.005
Prediabetes	Yes	0%	1%	8%	<0.001
Dyslipidemia	Yes	0%	5%	77%	<0.001
Arterial hypertension	Yes	1%	5%	18%	<0.001
History of weight loss	Yes	2%	36%	40%	<0.001
Gout	Yes	0%	0%	4%	<0.001
Metabolic syndrome	Yes	0%	3%	88%	<0.001
Post-insulin therapy weight gain	Yes	0%	0%	1%	0.19
Postpartum	Yes	0%	5%	5%	<0.001
Post-metabolic surgery	Yes	0%	0%	0%	0.15

Abbreviations: PMS—premenstrual syndrome; PCOS—polycystic ovarian syndrome; *p*-value—Pearson chi-square.

**Table 12 nutrients-16-03291-t012:** Specific conditions and health issues based on BFP classification.

Variable	Class	Normal FM (*n* = 394)	High FM (*n* = 225)	Very High FM (*n* = 542)	*p*-Value
Food intolerance	Yes	3%	2%	4%	0.33
PMS	Yes	3%	12%	11%	<0.001
Weight gain after quit smoking	Yes	0%	3%	3%	0.004
Oral contraceptives	Yes	2%	7%	4%	0.004
Menopause	Yes	8%	9%	12%	0.15
Preclimax	Yes	1%	8%	6%	<0.001
PCOS	Yes	1%	4%	3%	0.04
Prediabetes	Yes	0%	2%	8%	<0.001
Dyslipidemia	Yes	2%	34%	67%	<0.001
Arterial hypertension	Yes	2%	9%	18%	<0.001
History of weight loss	Yes	8%	40%	37%	<0.001
Gout	Yes	0%	2%	3%	0.006
Metabolic syndrome	Yes	2%	32%	79%	<0.001
Post-insulin therapy weight gain	Yes	0%	0%	1%	0.18
Postpartum	Yes	1%	6%	4%	0.002
Post-metabolic surgery	Yes	0%	0%	0%	0.38

Abbreviations: PMS—premenstrual syndrome; PCOS—polycystic ovarian syndrome; FM—fat mass; *p*-value—Pearson chi-square.

**Table 13 nutrients-16-03291-t013:** Multinomial logistic regression.

Predictors	Overweight OR (95% CI)	Overweight *p*-Value	Obese OR (95% CI)	Obese *p*-Value
Hydration status	0.76 (0.68–0.84)	**<0.001**	0.68 (0.60–0.76)	**<0.001**
Holiday [Yes]	0.04 (0.00–0.35)	**0.004**	0.02 (0.00–0.23)	**0.001**
Whole foods [Yes]	0.05 (0.01–0.21)	**<0.001**	0.04 (0.01–0.21)	**<0.001**
Age	0.92 (0.87–0.98)	**0.006**	0.97 (0.92–1.03)	0.359
Sleep hours	0.23 (0.12–0.45)	**<0.001**	0.23 (0.11–0.45)	**<0.001**
Fat tissue	1.18 (1.03–1.35)	**0.018**	1.24 (1.08–1.43)	**0.002**
FMH Hypertension [Yes]	11.69 (1.90–72.01)	**0.008**	16.01 (2.49–102.80)	**0.003**
Dessert [Yes]	1.39 (0.36–5.31)	0.632	16.42 (4.03–66.94)	**<0.001**
Cooked food [Yes]	0.07 (0.02–0.27)	**<0.001**	0.05 (0.01–0.23)	**<0.001**
Alcohol [Yes]	3.17 (0.50–20.15)	0.221	7.56 (1.14–50.29)	**0.036**
PCOS [Yes]	1.95 (1.08–3.53)	**0.027**	1.65 (0.91–2.98)	0.096
Prediabetes [Yes]	1.62 (0.64–4.10)	0.311	23.30 (9.18–59.13)	**<0.001**
Three meals/day [Yes]	0.13 (0.03–0.53)	**0.004**	0.01 (0.00–0.02)	**<0.001**
R^2^ Nagelkerke = **0.907**				

Abbreviations: OR—odds ratio; PCOS—polycystic ovarian syndrome; FMH—family medical history; *p*-value—Wald test.

**Table 14 nutrients-16-03291-t014:** Obesity risk factors based on BMI model.

Risk Factors	Odds Ratio	95% CI	*p*-Value
Age	1.08	1.03–1.13	**0.004**
Hydration status	0.84	0.75–0.92	**<0.001**
FMH Diabetes [Yes]	0.33	0.11–0.87	**0.032**
Craving [Yes]	17.08	4.82–68.55	**<0.001**
Fatigue [Yes]	16.90	5.20–64.03	**<0.001**
Stress [Yes]	42.07	12.70–172.12	**<0.001**
Nibbling [Yes]	4.37	1.19–17.04	**0.029**
Compulsive [Yes]	32.93	10.16–128.04	**<0.001**
Quickly/A lot [Yes]	0.18	0.04–0.76	**0.025**
Large portions [Yes]	0.06	0.01–0.40	**0.005**
Fast food [Yes]	10.20	3.54–31.95	**<0.001**
After meals [Yes]	26.98	8.29–106.49	**<0.001**
Restaurant [Yes]	0.12	0.03–0.49	**0.003**
Anger/Upset [Yes]	0.14	0.03–0.53	**0.006**
PMS [Yes]	0.10	0.03–0.38	**0.001**
Other devices [Yes]	0.15	0.03–0.79	**0.024**
R^2^ Nagelkerke = **0.959**			

Abbreviations: PMS—premenstrual syndrome; CI—confidence interval; *p*-value—Wald test.

**Table 15 nutrients-16-03291-t015:** Obesity risk factors based on the BFP model.

Risk Factors	Odds Ratio	CI	*p*-Value
BMI	1.09	1.03–1.18	**0.007**
Muscular tissue	0.68	0.62–0.75	**<0.001**
BMR	1.01	1.00–1.01	**<0.001**
Hydration status	0.86	0.77–0.95	**0.005**
FMH Diabetes [Yes]	1.66	1.01–2.76	**0.046**
Pleasure [Yes]	1.78	1.06–3.03	**0.031**
Quickly/A lot [Yes]	0.39	0.16–0.91	**0.032**
Anger/Upset [Yes]	2.72	1.22–6.10	**0.015**
PMS [Yes]	0.38	0.17–0.83	**0.017**
TV [Yes]	2.28	1.26–4.13	**0.006**
Daily fruits [Yes]	0.26	0.12–0.55	**<0.001**
Alcohol [Yes]	1.98	1.11–3.57	**0.022**
Smoking [Yes]	4.97	2.09–12.36	**<0.001**
Menopause [Yes]	0.16	0.07–0.35	**<0.001**
Preclimax [Yes]	0.18	0.06–0.51	**0.001**
History of weight loss [Yes]	0.47	0.28–0.78	**0.004**
≥30 min PA/day	3.33	1.67–6.84	**0.001**
R^2^ Nagelkerke = **0.831**			

Abbreviations: PMS—premenstrual syndrome; BMI—body mass index; BMR—basal metabolic rate; PA—physical activity; CI—confidence interval; *p*-value—Wald test.

**Table 16 nutrients-16-03291-t016:** Comparison between BMI and BFP models.

Model	AUC	95% CI	Accuracy	Specificity	Sensitivity
BMI	0.998	0.996–0.999	0.983	0.985	0.982
BFP	0.975	0.967–0.982	0.922	0.937	0.904

Abbreviations: BMI—body mass index; BFP—body fat percentage; AUC—area under ROC curve; CI—confidence interval.

## Data Availability

The datasets used and/or analyzed during the current study are available from the corresponding author upon reasonable request due to ethical reasons.

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
