# Peer review of "Comparative Analysis of Dietary Habits and Obesity Prediction: Body Mass Index versus Body Fat Percentage Classification Using Bioelectrical Impedance Analysis"

_nutrients, 2024, doi:10.3390/nu16193291_

Round 1
Reviewer 1 Report
Comments and Suggestions for Authors
The authors in their work address a very interesting topic of classification of overweight and obesity, however, there are many inaccuracies in the manuscript
1. In the introduction, the authors completely omitted indicators that are available in the literature on the subject, such as WHR, Sheldon's typology, Broca's formula with later modifications, Rohrer's index, leanness index, RBMI index, or FFF indicators that take into account measurements of adipose tissue, skinfold measurements, BAI index, MT (muscle-fat) index or BMIFat
2. The authors indicate that the basic cause of overweight or obesity is a positive energy balance resulting from improper nutrition, however, athletes eat much more than the basic BMR and are not obese, therefore the concept of positive energy balance should not be simplified as bad eating habits
3. In the methodology, the authors did not indicate what tools they used to assess physical activity, sleep or eating habits - were they standardized questionnaires?
4. The authors divided the groups according to BMI, dividing them according to the classification into normative body mass, overweight and obesity, unfortunately in the case of BFP the division is not made correctly, there is no classification into overweight or obesity and the division refers to fat mass, therefore the division should take into account the daily amount of FM, high and very high
5. The authors classified women and men in one group, but their FM level is different, I know that in the case of the division according to BFP these differences were taken into account, but after all the eating habits, lifestyle, physical activity undertaken are different for women and men and a division should be made according to gender.
6. Lifestyle, physical activity, occurrence of medical events in relatives from the closest family depends on age for a person aged 25 their parents may not have suffered a heart attack, stroke or been diagnosed with type 2 diabetes yet, while for the same person aged 45 such events may have already occurred. It is important to know whether a heart attack, hypertension, stroke, or type 2 diabetes occurred in members of the immediate family before the age of 45, the second limit is the age of 60. - such information should be included in research tools, hence my question about the standardization of surveys and questionnaires
7. Lack of information about the exclusion of people from the study group due to the inability to perform the test using the BIA method
8. Why did people report to the endocrinology clinic?
9. When describing statistical methods, the authors indicate the lack of normality of the distribution of the studied features in individual groups, why then were the means and not the medians and quartiles presented
10. The aim of the work was to indicate the influence of eating habits on the occurrence of overweight - this aim was not achieved, the influence cannot be assessed in a single study, we do not know what happened before, the assessment of habits was at a given moment at a given body weight. for the purpose of the work, the authors indicate that they want to compare the results from two classification models based on BMI and BFP - however, the results do not indicate a comparison - whether there are differences between these two classifications. To achieve this goal, it is necessary to compare the results of people classified as overweight according to BMI and high FM content according to BFP
11. In the results in Tables 1 and 2 there are no units, what does "0" mean, e.g. for milk?; why in the case of three "0" values, the authors indicate statistical significance at the level of p<0.001, if we show that there is a difference, the results must indicate it
12. In Tables 3-10, the authors show the answers 'YES', "NO" and yet their sum gives 100%, so one answer can be indicated and the tables will be much shorter.
BMI is a good method for assessing the nutritional status of young people up to about 45 years of age. In people entering the involutional ontogenetic stage, this indicator is disturbed by a change in body height. Many authors indicate that at the age of 60+, health well-being is associated with a BMI value of 26-28 kg/m2, and the authors did not take this into account
The work requires substantial methodological corrections
Author Response
Esteemed Reviewer,
Thank you for your thorough review and insightful comments. We have carefully considered each of your suggestions and have provided detailed responses and explanations in the attached document. The file includes our revisions, justifications for any decisions made, and how we addressed each of your points. We believe these revisions have improved the manuscript. Please do not hesitate to reach out if there are any additional concerns or points for further discussion.

Reviewer 2 Report
Comments and Suggestions for Authors
The study showed some interesting data. There seemed to be some suggested parts to be reconsidered.
1. Considering the overall results, why were the dietary habits seen in the title stressed? The reasons for stressing the dietary habits should be detailed more in Introduction and Discussion.
2. The novelty of dietary habits on obesity should be more stressed in the study.
3. The study could be done by gender because the body water and BFP level might differ by gender.
4. The prior studies’ results that compared the data between BMI and BFP could be summarized in Introduction.
5. Was the prevalence of normal, overweight and obese subjects as shown in Table 1 similar with that of national data? That is important in considering the representativeness of sampling.
6. As shown in Table 15, the difference in AUC between BMI and BFP was significant statistically; however, was it significant clinically? It might be discussed more.
7. The OR with 95%CI could be presented in the categories of normal (referent), overweight and obesity. This appeared essential since the earlier data were presented among three categories.
8. The AUC estimation might be in the same idea in the suggestion No. 7.
9. In recruiting the subjects, was a cancer history considered? The history may affect dietary habits and body composition.
10. In recruiting the subjects, was the history of psychic diseases such as depression considered? The history may affect dietary habits and body composition.
11. The accuracy and precision of Tanita analyzer should be described.
12. Row 163; was 18 of BMI right? 18 may be not normal. 18.5 can be often used in prior studies.
13. What was the principle to use the uppercase letter the expression of, for instance, ‘Inclusion Criteria’, ‘Eating habits’?
14. Chronic kidney disease (CKD) can affect the body water. Could the subjects with CKD be screened using a creatinine measure and excluded?
15. Severe and chronic liver disease, for instance cirrhosis, can affect the body water. Could the subjects with liver diseases be excluded?
16. Conclusions might be long. The sentences included the points as described in the study limitations.
17. Abstract; 95% CI could be added to OR.
18. Abstract; a space was necessary after Conclusions.
19. Row 39-40; this sentence could have an appropriately cited reference.
20. Row 19 and153; the decimal points of each variable should be unified appropriately in 36 and 11.90.
Comments on the Quality of English LanguageModerate editing of English language required. Please recheck.
Author Response
Esteemed Reviewer,
Thank you for your comprehensive review and thoughtful feedback. We have carefully evaluated each of your suggestions and provided detailed responses in the attached document. This file outlines the revisions we have made, the reasoning behind our decisions, and how we have addressed your concerns. We believe these changes have enhanced the manuscript. If you have any further questions or require additional clarification, please do not hesitate to contact us.

Round 2
Reviewer 1 Report
Comments and Suggestions for Authors
The authors partially followed the previous comments, but still did not compare the two assessment methods. The aim of the work was to determine which method is more precise and the authors presented whether people classified as having abnormal body mass according to BMI and BFP differ from each other. The work is further a summary of information on how the groups differ from each other when we divide them according to BMI and BFP, however, it is not known which, according to the authors, is better. Additionally, why was a control group not used in the case of division according to BFP
Author Response
Esteemed Reviewer,
Thank you for your insightful comment. While you suggested comparing the two evaluated models, BMI and BFP, we would like to clarify that these comparisons were conducted from the outset. The results are detailed on pages 20 and 21 of the manuscript and are also presented in Table 15 and Figure 2. Indeed, our research aimed to identify the more accurate classification method. As our statistical analysis indicates, BMI demonstrated superior accuracy with higher sensitivity and specificity: AUC of 0.998 (95% CI: 0.996 - 0.999) compared to the BFP model: AUC of 0.975 (95% CI: 0.967 - 0.982).
For both BMI and BFP models, the control group was utilized. The description of the three groups—control, overweight, and obesity—for the BMI model can be found in the Materials and Methods section, specifically on page 5 of the current manuscript. Additionally, following your suggestion from round 1 of comments, we adjusted the BFP model groups to normal fat mass, high-fat mass, and very high-fat mass. The normal fat mass group serves as the control group for the BFP model, detailed on page 5 as well.
If we have misunderstood your suggestion, we apologize and kindly ask you to provide further clarification so that we can make the necessary adjustments.
Kind regards,
Denisa Pescari
Reviewer 2 Report
Comments and Suggestions for Authors
The revisions improved the paper, while there were still reconsidered parts.
1. Gender difference in body composition is described in prior studies, while the authors’ study gender was not significantly associated with obesity. The reasons for the discrepancies across studies could be described more.
2. The OR with 95%CI should be presented in the categories of normal (referent), overweight and obesity. At least, the supplemental presentation would be necessary. Was the OR of overweight close to that of normal or obesity? Or was it neutral?
3. Those with kidney or liver damage were excluded; then, what was their damage? The levels of laboratory data and/or the name of diseases could be detailed.
Comments on the Quality of English Language
Recheck the overall text, please.
Author Response
Esteemed Reviewer,
Thank you for your comments. Below, you will find the file with our responses to your suggestions. The changes have been made and are reflected in the new manuscript.
Kind regards,
Denisa Pescari

Round 3
Reviewer 1 Report
Comments and Suggestions for Authors
We still do not understand each other. Throughout the manuscript, the authors show separate classifications according to BMI and BFP, and the title is a comparison of these classifications. In order to compare something, the patient no. 1 must be classified according to both BMI and BFP and see according to which classification we get more possibilities. On pages 20 and 21 indicated by the authors there is information about risk factors and here again there is a division into groups according to BMI and separately BFP
Author Response
Esteemed Reviewer,
We greatly appreciate your suggestion regarding the comparison of the two evaluated models. Your requests concerning the classification of participants have been implemented. Thus, the nutritional status of each participant from the total sample of 1,255 subjects was classified both in terms of BMI value and body fat percentage severity. To avoid confusion regarding the page numbering (which was redone after the manuscript was uploaded following the suggestions of Reviewer 2), the results of this statistical method are detailed in Table 16. We cannot present these results in a table with 1,255 individuals, but if possible, we can provide the database. Therefore, the BMI and BFP models were subsequently compared following the classification of participants as described in the Methods section. After the participants were classified as mentioned earlier and as indicated in the Methods section, the two models were compared in terms of accuracy, specificity, and sensitivity (Figure 2). In conclusion, your requests have been addressed since the submission of the first version of the manuscript regarding this suggestion, and from a statistical standpoint, no further action can be taken.
Sincerely,
Denisa Pescari
Reviewer 2 Report
Comments and Suggestions for Authors
Most parts were revised. If it is possible, the OR with 95%CI as presented in the categories of normal, overweight and obesity may give more information to readers.
Author Response
Esteemed reviewer,
Thank you for your valuable feedback requesting odds ratios (ORs) for both overweight and obesity. While our initial study design was focused on identifying factors specifically associated with obesity, your suggestion prompted us to create an additional model that includes both overweight and obesity as distinct outcomes in the analysis. This new model allowed us to identify and present both risk and protective factors for each weight category, expanding the scope of our study. Although this extended analysis was not part of our original approach, we found that it revealed significant and relevant results for overweight as well. These findings provided a more comprehensive understanding of the factors influencing weight status and added further depth to our analysis. We appreciate your suggestion, as this expanded approach strengthens the study and provides more detailed insights into the predictors for both overweight and obesity.
All these changes could be observed between 380-386 lines and 710-822 lines in revised version.